# Arctigenin Enhances the Cytotoxic Effect of Doxorubicin in MDA-MB-231 Breast Cancer Cells

**DOI:** 10.3390/ijms21082997

**Published:** 2020-04-23

**Authors:** Kyu-Shik Lee, Min-Gu Lee, Yun-Suk Kwon, Kyung-Soo Nam

**Affiliations:** Department of Pharmacology, School of Medicine and Intractable Disease Research Center, Dongguk University, Gyeongju 38066, Korea; there1@dongguk.ac.kr (K.-S.L.); mklee@dongguk.ac.kr (M.-G.L.); ys0727@dongguk.ac.kr (Y.-S.K.)

**Keywords:** arctigenin, doxorubicin, triple-negative breast cancer, apoptosis-inducing factor, cell death

## Abstract

Several reports have described the anti-cancer activity of arctigenin, a lignan extracted from *Arctium lappa* L. Here, we investigated the effect of arctigenin (ATG) on doxorubicin (DOX)-induced cell death using MDA-MB-231 human breast cancer cells. The results showed that DOX-induced cell death was enhanced by ATG/DOX co-treatment in a concentration-dependent manner and that this was associated with increased DOX uptake and the suppression of multidrug resistance-associated protein 1 (MRP1) gene expression in MDA-MB-231 cells. ATG enhanced DOX-induced DNA damage and decreased the phosphorylation of signal transducer and activator of transcription 3 (STAT3) and the expressions of RAD51 and survivin. Cell death caused by ATG/DOX co-treatment was mediated by the nuclear translocation of apoptosis inducing factor (AIF), reductions in cellular and mitochondrial Bcl-2 and Bcl-xL, and increases in mitochondrial BAX levels. However, caspase-3 and -7 did not participate in DOX/ATG-induced cell death. We also found that DOX/ATG-induced cell death was linked with activation of the p38 signaling pathway and suppressions of the phosphorylations and expressions of Akt and c-Jun N-terminal kinase. Taken together, these results show that ATG enhances the cytotoxic activity of DOX in MDA-MB-231 human breast cancer cells by inducing prolonged p21 expression and p38-mediated AIF-dependent cell death. In conclusion, our findings suggest that ATG might alleviate the side effects and improve the therapeutic efficacy of DOX.

## 1. Introduction

Breast cancer is the major cause of cancer-related death among women. Although many subtypes of breast cancer have been reported, they are generally classified as hormone receptor (HR)-positive, human epidermal growth factor receptor 2 (HER2)-overexpressing, and triple-negative breast cancer (TNBC) [1]. Triple-negative breast cancer (TNBC) has a higher mortality rate than HR-positive or HER-2-overexpressing breast cancer because of its high rate of recurrence [2,3,4]. Furthermore, because of the absence of HR and HER-2 in TNBC, non-targeting anti-cancer drugs such as paclitaxel, cyclophosphamide, and doxorubicin (DOX) are being used to treat the disease [5].

DOX is an anthracycline antibiotic and broad spectrum anti-cancer agent [6,7]. Although DOX is useful for the treatment of triple-negative breast cancer (TNBC), in practice its use is limited because of its serious side effects, which include cardiotoxicity, diarrhea, vomiting, hair loss, and nausea. Actually, the total dose of DOX administered must not exceed 450–500 mg/m^2^ because of its cardiotoxicity [6]. Furthermore, reductions in DOX dosage made to address its side effects reduce its therapeutic efficacy [8]. 

Arctigenin (ATG) is a pharmaceutically active substance isolated from the seeds of *Arctium lappa* L. (commonly called greater burdock), and several investigators have shown it has anti-viral, anti-inflammatory, anti-cancer, and immunomodulatory activities [9,10,11,12,13]. The anti-cancer activity of ATG has been reported to due to the induction of apoptosis mediated by mitochondrial disruption and cell cycle arrest in breast, lung, bladder, gastric, hepatic, and colon cancer cells [14,15,16,17,18]. In a recent study, we showed ATG suppressed metastatic potential and induced autophagic cell death by inhibiting estrogen receptor (ER) expression in MCF-7 human breast cancer cells [19,20]. Also, Wang et al. reported human non-small cell lung cancer (NSCLC) cells treated with ATG exhibited greater chemosensitivity to cisplatin-induced apoptotic cell death mediated by the down-regulation of survivin [21].

Combination chemotherapies are being increasingly used to treat cancers to minimize toxicities and side effects based on the delivery of lower doses of the drugs responsible [22,23]. Numerous investigations have shown ATG has anti-cancer and anti-metastatic effects on different cancer cell types. Therefore, we assessed the effects of ATG/DOX co-treatment to determine whether ATG enhances the cytotoxic effect of DOX in MDA-MB-231 TNBC cells.

## 2. Results

### 2.1. ATG Enhanced DOX-Induced MDA-MB-231 Cell Death 

We evaluated whether DOX cytotoxicity was enhanced by ATG in MDA-MB-231 cells. When MDA-MB-231 cells were treated with 0.2 μM DOX for 72 h, cell viability reduced to 72%, but combined treatment with 0.2 μM DOX and ATG (10–200 μM) reduced viability to below 50% and ATG co-treatment reduced viability in a concentration-dependent manner (Figure 1A,B).

Moreover, Combination indices (CI) values quantitatively validated by Compusyn software was <1, indicating that ATG synergistically enhanced cytotoxicity of DOX (Figure 1C). The results imply that ATG is a potent substance for combinational treatment with DOX in breast cancer.

### 2.2. DOX Uptake by MDA-MB-231 Cells Was Increased by ATG

Next, we assessed intracellular DOX levels in MDA-MB-231 cells co-treated with ATG and DOX. We observed ATG co-treatment increased DOX uptake by cells (Figure 2A). Furthermore, ATG co-treatment increased DOX-induced H2A histone family member X (H2A.X) phosphorylation, decreased signal transducer and activator of transcription 3 (STAT3) phosphorylation and expression, and down-regulated survivin and DNA repair protein RAD51 homolog 1 isoform 1 (RAD 51) protein expressions (Figure 2B). In addition, we evaluated changes in the gene expression of ATP-binding cassette (ABC) transporters multidrug resistance-associated protein 1 (MRP1) and breast cancer resistance protein 1 (BCRP), because the effectiveness of chemotherapy is negatively associated with the expressions of these factors [24]. We found that ATG co-treatment reduced the gene expression of MRP1 but did not affect the gene expression of BCRP (Figure 2C). This result suggests that augmentation of DOX cytotoxicity by ATG is mediated by enhancing DNA damage and suppressing DNA repair by increasing DOX uptake and reducing MRP1 transcription.

### 2.3. Cell Death by ATG/DOX Co-Treatment Was Associated with Down-Regulations in Bcl-2 and Bcl-xL and Increases in BAX Levels in Mitochondria

ATG co-treatment accelerated DOX-induced cell death (Figure 1B). Furthermore, survivin (an anti-apoptotic factor) induction by DOX was concentration-dependently suppressed by ATG (Figure 2B), implying apoptotic signaling was activated by ATG/DOX co-treatment [25,26]. In addition, we found that B-cell lymphoma 2 (Bcl-2) and B-cell lymphoma-extra-large (Bcl-xL) levels in whole cell lysates and mitochondrial fractions were diminished by ATG/DOX co-treatment and that mitochondrial Bcl-2-associated X protein (BAX) levels were increased (Figure 3). In contrast, ATG/DOX co-treatment did not induce the cleavages of caspase-3 and -7 (Figure 4A). Furthermore, the enhancement of cell death by ATG was not associated with receptor interacting serine/thereonine kinase 3 (RIPK3)-mediated necroptosis or Beclin 1-linked autophagy (Figure 4B). These results show cell death enhancement by ATG/DOX co-treatment was not mediated by caspase-3-dependent apoptosis, RIP3K-mediated necroptosis, or Beclin 1-linked autophagy.

### 2.4. MDA-MB-231 Cell Death by ATG/DOX Co-Treatment Was Induced AIF-Dependently

Survivin prevents caspase-independent apoptosis inducing factor (AIF)-dependent cell death [26]. Accordingly, we evaluated the effect of ATG/DOX co-treatment on the nuclear translocation of AIF to determine whether inhibition of survivin expression by ATG/DOX co-treatment was associated with AIF-dependent cell death. The results showed that nuclear AIF levels were increased by ATG/DOX co-treatment as determined by Western blotting and immunocytochemistry (Figure 5A,B), which indicated ATG/DOX co-treatment-induced cell death was mediated by AIF.

### 2.5. Sustained Increase of p21 Was Associated with ATG/DOX Co-Treatment-Induced Cell Death

p21 is a cyclin-dependent kinase that induces cell cycle arrest [27] and has been reported to be involved in cell death [28,29]. Therefore, we evaluated the effect of ATG/DOX co-treatment on p21 expression. The mRNA and protein levels of p21 were enhanced by ATG/DOX co-treatment in a dose-dependent manner (Figure 6A,B). Furthermore, the up-regulation of its mRNA and protein levels were sustained at 24–72 h after ATG/DOX co-treatment, which suggested ATG/DOX co-treatment-induced cell death involved the up-regulation of p21.

### 2.6. Involvement of p38 MAPK Phosphorylation in ATG/DOX Induced Cell Death

C-Jun N-terminal kinase (JNK), extracellular signal-regulated kinase (ERK), and p38, and Akt play important roles in cell survival and death [30,31]. Hence, we evaluated whether ATG/DOX co-treatment altered the phosphorylations of MAPKs and Akt. We found p38 MAPK phosphorylation was increased by co-treatment and that this co-occurred with the nuclear translocation of AIF and reductions in the phosphorylations of Akt and JNK (Figure 7). Furthermore, ribosomal protein S6 phosphorylation induced by DOX was prevented by ATG/DOX (Figure 7), whereas DOX-induced phosphorylations of ERK and Rel/p65 were unaffected (Figure 7). These results suggest increased cell death by ATG/DOX co-treatment was mediated by the activation of p38 and by suppressions of the JNK and Akt signaling pathways.

## 3. Discussion

Although DOX is useful anti-cancer drug, its side effects severely restrict its use. However, reducing DOX dosages markedly mitigate these side effects. In the present study, we found the cytotoxic activity of DOX was significantly enhanced by co-treating it with ATG in MDA-MB-231 human triple-negative breast cancer cells and that DOX uptake was dose-dependently enhanced and MRP1 expression was dose-dependently suppressed in these cells (Figure 1B and Figure 2A,C). Furthermore, the presence of ATG synergistically increased DOX cytotoxicity (Figure 1C). These results suggest that combinatorial ATG/DOX should be considered a potential treatment for triple-negative breast cancer that reduces the side effects of DOX.

In the present study, ATG co-treatment increased DOX-induced H2A.X phosphorylation, reversed DOX-induced survivin and RAD51 protein expressions, and increased DOX uptake by MDA-MB-231 cells (Figure 2A,B). H2A.X phosphorylation is the result of DNA damage [32,33], and thus, we speculated that increased DOX-induced H2A.X phosphorylation by ATG reflects an increase in DNA, and that this enhanced MDA-MB-231 cell death (Figure 1C). Cell death by DNA damage is primarily caused by a failure to repair DNA [34,35], and survivin and RAD51 play important roles in DNA repair [36,37]. Hence, the ATG/DOX-induced suppressions of survivin and RAD51 protein levels suggests increased cell death was due to an inability to repair DNA. Consequently, the present results suggest that ATG/DOX co-treatment-induced MDA-MB-231 cell death was associated with DNA damage and failure of the DNA repair system.

Constitutively activated STAT3 has been observed in various cancers and shown to enhance cell proliferation, invasion, and survival and to inhibit apoptosis in breast cancer cells [38]. Studies have also shown the growth of breast cancer cells is inhibited by STAT3 inhibitors. Apoptotic cell death was observed by treating MDA-MB-231 human breast cancer cells with STAT3-siRNA or inhibiting STAT3, for example, decreased phosphorylation of STAT3 by hydrazinocurcumin was associated with down-regulations of the expressions anti-apoptotic and metastasis-enhancing factors [39,40,41]. Furthermore, survivin expression is transcriptionally regulated by STAT3 [42]. In the present study, STAT3 phosphorylation and survivin were down-regulated by ATG/DOX co-treatment in MDA-MB-231 cells (Figure 2B). These results suggest that ATG/DOX co-treatment enhances cell death by down-regulating DOX-induced survivin expression and STAT3 phosphorylation. 

Decreases in Bcl-xL and Bcl-2 levels are closely associated with apoptotic cell death mediated by mitochondrial disruption [43]. In contrast, increases in BAX (a member of the pro-apoptotic Bcl-2 family) levels in mitochondria is an indicator of apoptotic cell death [44]. In the present study, we observed ATG/DOX co-treatment reduced cellular and mitochondrial Bcl-xL and Bcl-2 levels and increased mitochondrial BAX levels (Figure 3), which suggests ATG/DOX co-treatment induced cell death by disrupting mitochondrial integrity by down-regulating Bcl-xL and Bcl-2 levels and inducing the translocation of BAX to mitochondria.

Caspase-dependent apoptosis is a well-known type of programmed cell death system. However, in the present study, ATG/DOX co-treatment was not associated with the activations of caspase-3 or -7 (Figure 4A) and increases of LC-3II (an autophagy marker) and of RIP3K (a necrosis marker) were not observed (Figure 4B). In contrast, ATG was found to dose-dependently augment AIF protein levels as determined by a nuclear fractionation assay and fluorescence immunocytochemistry (Figure 5). AIF is a key inducer of caspase-independent cell death and causes chromatin condensation and DNA fragmentation [45]. In addition, we previously observed the nuclear translocation of AIF from mitochondria during apoptosis induced by staurosporin, etoposide, ceramide, or cisplatin [45,46]. Furthermore, it has been reported induction of Bcl-2 expression by epithelium-derived factor inhibited apoptotic cell death by suppressing the nuclear translocation of AIF from mitochondria [47], and that the up-regulation of mitochondrial BAX caused release of AIF from mitochondria by inducing mitochondrial outer membrane permeabilization [48]. In the present study, we observed ATG/DOX co-treatment down-regulated Bcl-2 and Bcl-xL and up-regulated mitochondrial BAX and nuclear AIF levels (Figure 3 and Figure 5). Consequently, our study indicates ATG/DOX co-treatment-induced MDA-MB-231 cell death was mediated by the nuclear translocation of AIF induced by mitochondrial damage.

Mammalian target of rapamycin (mTOR) complex 1/S6K1 signaling influenced T cell activation and differentiation without requiring S6 phosphorylation, which suggested S6 may affect cell responses independently of mTOR [49]. Furthermore, other investigators have reported DNA damage by 7,12-dimethylbenz(a)anthracene was prevented by S6 phosphorylation in a knock-out mouse model and that hyperphosphorylation of S6 was linked with poor prognosis in NSCLC and with metastatic potential in H1650 and SK-MES-1 NSCLC cells [50,51]. In the present study, we found that ATG co-treatment reversed DOX-mediated phosphorylation of S6 and the increase of survivin and RAD51 (Figure 2C and Figure 7). These results suggest ATG promotes DOX-mediated cell death by inhibiting DNA repair and improves prognoses by suppressing DOX-induced S6 phosphorylation.

p21 enhancement is primarily caused by the induction of cell cycle arrest and its expression is associated with p53 induced by DNA damage [52]. Many studies have shown that cell senescence is associated with p21 accumulation [53]. Moreover, the induction of apoptosis by suberoylanilide hydroxamic acid (a histone deacetylase inhibitor) was found to be induced by p21 overexpression in T24 human bladder carcinoma cells [54]. In the present study, we found ATG/DOX co-treatment resulted in the prolonged induction of p21 in MDA-MB-231 cells (Figure 6). Li et al. showed that oridonin-induced cell death by autophagy and apoptosis was also closely linked with the sustained up-regulation of p21 [29], and others have shown sustained p21 induction triggers intracellular reactive oxygen species (ROS) production [28,55]. It has been well established that DOX generated ROS has cardiotoxic effects in cancer patients. In the present study, we found that DOX uptake by MDA-MB-231 cells and the cytotoxic activity of DOX were increased by ATG (Figure 1 and Figure 2A). Consequently, these observations indicate that increased cytotoxic activity of DOX by ATG might be associated with enhanced ROS production mediated by the overexpression of p21. 

Phosphorylation of p38 during cell death has been reported on a number of occasions [56,57,58,59,60]. Furthermore, several research teams have shown p38 activation is linked with caspase-independent cell death mediated by the nuclear translocation of AIF [56,59,61,62]. Others have reported the mitochondrial translocation of BAX is induced by the phosphorylation of p38 [63,64]. In the present study, we observed that DOX-induced p38 phosphorylation was dose-dependently enhanced by ATG/DOX co-treatment and that this matched increases in the nuclear and cytosolic levels of AIF (Figure 5 and Figure 6A). Moreover, ATG/DOX co-treatment was found to increase of BAX levels in mitochondria (Figure 3). Consequently, these results indicate that enhanced cell death by ATG/DOX co-treatment was caused by the p38-linked nuclear translocation of AIF and the mitochondrial translocation of BAX.

In the present investigation, we confirmed that ATG enhances the cytotoxic activity of DOX. This is consistent with the report that ATG increased cisplatin sensitivity in non-small cell lung cancer cells [21]. Furthermore, several studies showed that ATG suppresses chemical-induced metastatic potential in breast cancer cells, induces cell death of triple-negative breast cancer cells and has no significant toxicity less than 6 mg/kg in beagle dongs [20,65,66]. Anticancer drugs, such as doxorubicin, cisplatin, etc., have various side effects. To minimize the side effects, reducing the dosage of anticancer drug is an effective strategy. Therefore, the results suggest that ATG is a useful candidate to enhance therapeutic efficacy of anticancer drugs with diminished side effects.

Taken together, the present study demonstrates that ATG enhances the cytotoxic effect of DOX by increasing the cellular uptake of DOX and inducing the mitochondrial translocation of BAX, prolonged p21 induction, and the activation of p38, thus causing AIF-dependent cell death signaling. In conclusion, this investigation shows ATG should be considered a powerful candidate for combinational chemotherapy. However, this study proved the effect of ATG/DOX co-treatment at the cellular level. Therefore, further investigations in animal level must be required to evaluate the effect clearly.

## 4. Materials and Methods

### 4.1. Materials

ATG, DOX, and MTT reagent were purchased from Sigma Aldrich (Merck KGaA, Darmstadt, Germany). Tris base, glycine, and sodium chloride (NaCl) were obtained from BioShop Canada Inc. (Burlington, ON, Canada) and Dulbecco’s Modified Eagle’s medium (DMEM), antimycotic/antibiotic solution and trypsin were from Welgene, Inc. (Gyeongsan, Korea). Dimethyl sulfate (DMSO) was bought from Duksan Pure Chemicals (Ansan, Korea) and 30% acrylamide/bis-acrylamide solution from SERVA Electrophoresis GmbH (Heidelberg, Germany). Bicinchoninic acid (BCA) protein assay kits and horseradish peroxidase-conjugated goat anti-mouse and -rabbit IgG were obtained from Pierce Biotechnology (Rockford, IL, USA). Alexa 488 (Liedtke et al.)-conjugated goat anti-rabbit antibody was acquired from Thermo Fisher Scientific (Waltham, MA, USA), and antifade mounting solution containing 4′,6-diamidino-2-phenylindole (DAPI) was from BioLegend (San Diego, CA, USA). Sodium dodecyl sulfate (SDS) and *N,N,N′,N′*-tetramethylethylenediamine were purchased from VWR Life Science AMRESCO biochemicals (Solon, OH, USA). Antibodies for H2A.X, phospho-H2A.X, STAT3, phospho-STAT3, survivin, RAD51, Bcl-2, Bcl-xL, BAX, AIF, PARP, caspase-3, caspase-7, RIPK3, LC3, Beclin 1, p38, phospho-p38, Akt, phospho-Akt, JNK, phospho-JNK, S6, phospho-S6, ERK, phospho-ERK, Rel/p65, phospho- Rel/p65, mitochondrial cytochrome c oxidase subunit IV (COX IV), Lamin B1 and glyceraldehyde-3-phosphate dehydrogenase (GAPDH) were purchased from Cell Signaling Technology (Beverly, MA, USA). Antibody for β-actin was from Santa Cruz Biotechnology Inc. (Dallas, TX, USA).

### 4.2. Cell Culture

MDA-MB-231 human breast cancer cells were obtained from the Korean Cell Line Bank (Seoul, Korea) and grown in DMEM containing 1% antimycotic/antibiotic solution (100 units/mL of penicillin, 100 µg/mL of Streptomycin and 0.25 µg/mL amphotericin B) and 10% heat-inactivated fetal bovine serum (FBS; American Type Culture Collection, Manassas, VA, USA) in a 5% CO_2_ atmosphere at 37 ˚C. To perform experiments, culture media were replaced with conditioned media supplemented with 1% antimycotic/antibiotic solution and 2% FBS.

### 4.3. Cell Viability Analysis

MTT assay was performed to evaluate cytotoxicity of DOX and ATG and to assess whether ATG increases cytotoxic activity of DOX in MDA-MB-231 cells. MDA-MB-231 cells (5 × 10^3^) were plated in the wells of a 96-well plate and allowed to attach for 24 h. Culture medium was then removed and conditioned medium containing DOX and ATG was added. After co-culture for 72 h, 20 μL of 5 mg/mL MTT reagent was added, and cells were incubated for 4 h in a dark. Media were then removed and the formazan was dissolved by adding DMSO. Optical densities were measured at 540 nm using a Spectramax M2 spectrophotometer (Molecular Devices, LLC, Sunnyvale, CA, USA).

### 4.4. Doxorubicin Uptake Assay

We performed flow cytometric analysis to investigate the effect of ATG on DOX uptake in MDA-MB-231 cells. MDA-MB-231 cells were seeded into 6-well plates at 2 × 10^5^ cells/well and allowed to attached for 24 h. Cells were then pre-treated with 0 to 200 μM of ATG for 48 h and administrated with 0.2 μM DOX for 24 h, detached by trypsinization, washed twice with phosphate buffered saline (PBS) containing 2% FBS, and resuspended in PBS containing 2% FBS. Resuspended cells were analyzed by FACSCalibur II flow cytometry (Becton Dickinson Biosciences, San Jose, CA, USA).

### 4.5. Determining Combined Drug Interactions

Combination index (CI) values were calculated to determine whether 10–200 μM ATG synergistically enhanced the cytotoxicity of DOX. The association between the effects of ATG- and DOX-alone and in combination were analyzed using Compusyn Version 1.0 software (Combosyn Inc., Paramus, NJ, USA), as previously described [19]. CI values were determined for each dose and the corresponding effect level, presented as the fraction affected (Fa). The CI values at different Fa levels was automatically simulated as CI-Fa plot by Compusyn software. The CI values obtain a quantitative definition for the synergism (CI < 1), additive effect (CI = 1) and antagonism (CI > 1) of ATG/DOX combinations.

### 4.6. Quantitative Real-Time Polymerase Chain Reaction

To estimate the change of gene expression in MRP1 and BCRP1, quantitative real-time polymerase chain reaction was conducted. MDA-MB-231 cells (2 × 10^5^ cells/well) were plated into 6-well plates and allowed to attach for 24 h. Cells were then treated with 0.2 μM DOX with ATG (0–200 μM) for 24 h in DMEM supplemented with 2% FBS. The cells were harvested by trypsinization and lysed using the easy-BLUE^TM^ Total RNA Extraction Kit (iNtRON Biotechnology, Inc., Sungnam, Korea). Total RNA concentrations were measured using a NanoDrop spectrophotometer (Schimadzu Scientific Instruments, Columbia, MD, USA) and cDNA synthesis was performed using 1 µg of total RNA and Goscript^TM^ Reverse Transcriptase (Promega, Madison, WI, USA). Relative expressions of MRP1 and BCRP1 were determined by real-time PCR using QGreen 2× SybrGreen Master Mix (Cellsafe, Suwon, Korea) and an Eco^TM^ Real-time PCR unit (Illumina, San Diego, CA, USA). PCR products were confirmed by melting curve analysis and relative expressions were evaluated using EcoStudy v. 5.0.4890 software (Illumina). GAPDH was used as the internal control. PCR primers were synthesized by Bioneer Corporation (Daejeon, Korea) and the primer sequences of target genes were as follows: MRP1, forward 5′-GCGAGTGTCTCCCTCAAACG-3′ and reverse 5′-TCCTCACGGTGATGCTGTTC-3′; BCRP1, forward 5’-GCAGATGCCTTCTTCGTTATG-3′ and reverse 5’-TCTTCGCCAGTACATGTTGC-3′; GAPDH, forward 5’-CTGCTCCTCCTGTTCGACAGT-3′ and reverse 5’-CCGTTGACTCCGACCTT CAC-3′.

### 4.7. Mitochondrial Fractionation

Subcellular localization of Bcl-xL, Bcl-2, and BAX proteins were observed by mitochondrial fractionation. After incubating MDA-MB-231 cells with 0.2 μM of DOX plus 0 to 200 μM of ATG in DMEM medium supplemented with 2% FBS, cells were collected by scraping with a rubber policeman, and floating cells were centrifuged at 1000 rpm for 5 min. Mitochondrial fractionation was performed using the Mitochondria/Cytosol Fractionation Kit (BioVision Inc., Milpitas, CA, USA). Prepared mitochondrial and cytosolic fractions were stored at −80 ˚C until required.

### 4.8. Nuclear Fractionation

We performed nuclear fractionation to assess nuclear AIF level. MDA-MB-231 cells were co-incubated with 0.2 μM of DOX plus 0–200 μM of ATG in DMEM medium supplemented with 2% FBS for 72 h, collected by scraping, and floating cells were obtained by centrifugation at 1000 rpm for 10 min. Cells were then washed with ice-cold PBS, resuspended in a hypotonic buffer (20 mM Tris-HCl pH 7.4, 10 mM NaCl, and 3 mM MgCl_2_) containing protease inhibitor cocktail, and held on ice for 15 min. NP-40 solution (1/8 vol., 10%) was then added to cell suspensions and vortexed for 10 s. Mixtures were then incubated on ice for 10 min and centrifuged at 3000 rpm for 10 min at 4 ˚C. Supernatants were collected (cytosolic fractions), and pellets were lysed with Cell Extraction Buffer (Invitrogen, Carlsbad, CA, USA) containing protease inhibitor cocktail for 30 min on ice, centrifuged at 14,000× g for 30 min at 4 ˚C, and supernatants (nuclear fractions) were transferred to new tubes. The nuclear fractions were used for assessing nuclear level of AIF by Western blotting.

### 4.9. Western Blotting

MDA-MB-231 cells were seeded into 6-well plates and allowed to attach for 24 h in culture medium, and then co-incubated with 0.2 μM of DOX and 0 to 200 μM of ATG for 72 h in DMEM medium supplemented with 2% FBS. Cells were then lysed with radioimmunoprecipitation assay lysis buffer (Biosesang, Seongnam, Korea) supplemented with protease inhibitor cocktail and phosphatase inhibitor cocktail (GenDEPOT, LLC, Barker, TX, USA) and centrifuged at 13,000 rpm for 10 min at 4 ˚C. Supernatants were removed (whole cell lysates) and stored at −22 ˚C until required. Total protein concentrations in whole cell lysates were determined using the BCA method. Same amounts of proteins were subjected to SDS-polyacrylamide gel electrophoresis (PAGE) on 8%–15% gel and transferred to polyvinylidenefluoride (PVDF) membranes (Pall Life Science, Port Washington, NY, USA), which were blocked with 1% bovine serum albumin (BSA) or 5% non-fat dry milk (Santa Cruz Biotechnology, Inc.) in Tris-buffered saline-Tween (TBS-T, 50 mM Tris-HCl, 150 mM NaCl, and 0.1% Tween-20) and probed with primary antibodies diluted at 1:3000 in 1% BSA or 5% non-fat dry milk in TBS-T overnight at 4 ˚C. Membranes were washed three times with TBS and treated with secondary antibody diluted at 1:5,000 in TBS for 1 h at room temperature. Target protein bands were developed using a chemiluminescent substrate and photographed using Luminescent Image Analyzer LAS-4000 (Fujifilm Corporation, Tokyo). Densities of target protein bands were analyzed using Scion Image software (Alpha 4.0.3.2) (Scion Corporation, Frederick, MD, USA).

### 4.10. Fluorescence Immunocytochemistry

The nuclear localization of AIF in MDA-MB-231 cells was observed by fluorescence immunocytochemistry. Cover slips were sterilized with 70% ethanol in PBS and by exposure to UV radiation for 10 min and then collagen-coated. Cells were plated on cover slips, cultured for 24 h, co-treated with DOX and 0–200 μM of ATG for 72 h, and serially fixed in ice cold methanol for 4 min and then in acetone for 2 min. After fixation, cells were blocked in phosphate buffered saline (PBS) supplemented with 10% FBS, probed with 1:200 diluted AIF antibody in PBS overnight at 4˚C, and treated with 1:200 diluted Alexa 488-conjugated goat anti-rabbit IgG in PBS for 2 h at room temperature in the dark. Cells were then mounted onto coverslips, dipped into antifade mounting solution containing DAPI and placed on glass slides. Alexa 488 and DAPI stained cells were observed and photographed under a fluorescence microscope (Carl Zeiss, Jena, Germany).

### 4.11. Statistical Analysis

MTT assay, real-time PCR, and Western blotting results were analyzed by one-way ANOVA followed by Dunnett’s post-hoc test using SPSS Ver. 20.0 software (SPSS, Inc., Chicago, IL, USA). Results are presented as means ± SDs and statistical significance was accepted for *p* values < 0.05.

## Figures and Tables

**Figure 1 ijms-21-02997-f001:**
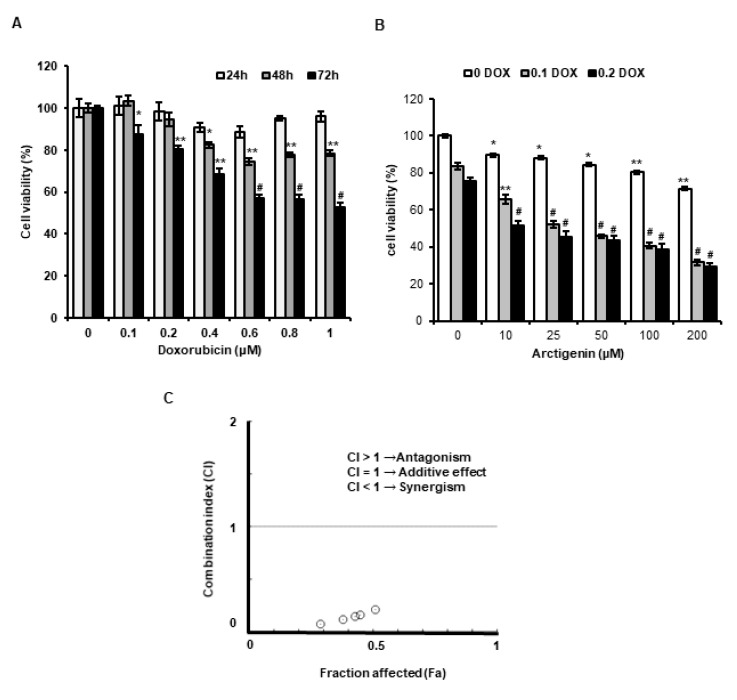
Effect of arctigenin (ATG) co-treatment on doxorubicin (DOX)-induced cytotoxicity in MDA-MB-231 cells. (**A**) Cells were incubated in Dulbecco’s Modified Eagle’s medium (DMEM) medium containing various concentrations of DOX (0–1 µM) for 24, 48, or 72 h. *, ** and # indicate *p* < 0.05, *p* < 0.01 and *p* < 0.001 vs. non-treated controls. (**B**) Cells were incubated in DMEM medium containing various concentration of ATG (0–200 μM) with or without 0.2 μM DOX for 72 h. ATG enhanced cytotoxicity of DOX in a concentration-dependent manner. * and ** indicate *p* < 0.05 and *p* < 0.01 vs. non-treated controls. ^##^ and ^###^ indicate *p* < 0.0005 and *p* < 0.0001 vs. non-treated controls. (A,B) Cell viabilities were determined using an MTT assay. All experiments were performed independently three times and results are presented as means ± SDs. (**C**) Combination indices (CI) versus fractional affected (Fa) plots for ATG/DOX co-treatment were graphically represented by Compusyn software. Synergistic cytotoxic activity of ATG/DOX co-treatment was observed in MDA-MB-231 human triple negative breast cancer cells. A CI value of < 1 indicates a synergistic cytotoxic effect.

**Figure 2 ijms-21-02997-f002:**
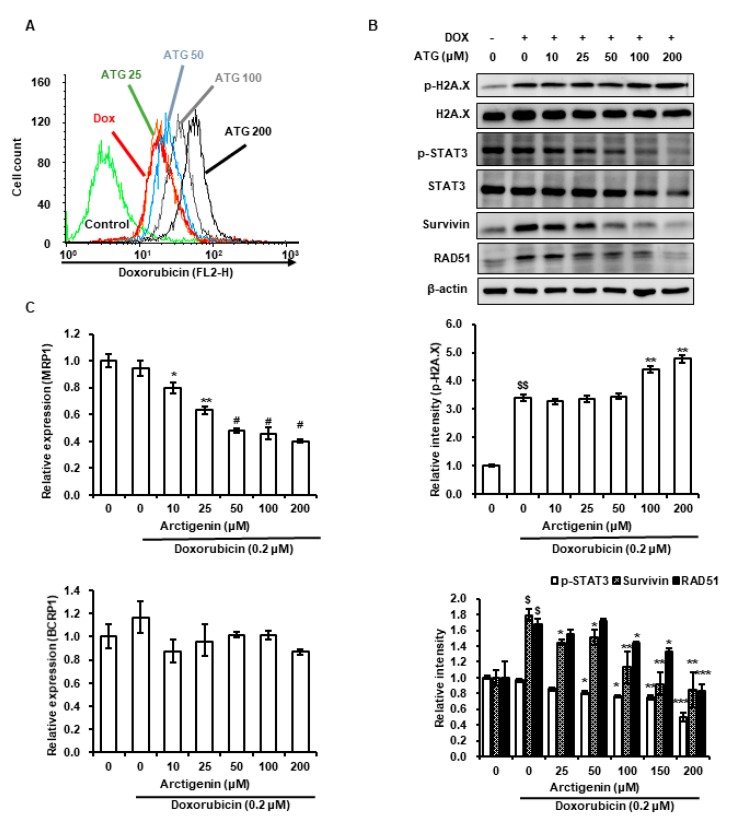
Effects of ATG on DOX uptake, the transcriptions of multidrug resistance-associated protein 1 (MRP1) and breast cancer resistance protein 1 (BCRP1), the phosphorylations of H2A histone family member X (H2A.X) and signal transducer and activator of transcription 3 (STAT3), and the expressions of survivin and DNA repair protein RAD51 homolog 1 isoform 1 (RAD51) in MDA-MB-231 cells. (**A**) Cells were grown for 48 h with various concentrations of ATG (0–200 μM) prior to being treated with 0.2 μM DOX treatment for 24 h. DOX uptake was analyzed by flow cytometry. ATG accelerated DOX uptake in a concentration-dependent manner. The X-axis shows fluorescence intensities of intracellular DOX and the Y-axis cell numbers per channel. (**B**) Cells were attached for 24 h and further grown for 24 h in DMEM medium supplemented 2% fetal bovine serum (FBS), 0.2 μM DOX, and various concentration of ATG (0–200 μM). MRP1 gene expression was suppressed by ATG/DOX co-treatment. Relative expressions of MRP1 and BCRP1 genes were evaluated in triplicate and normalized versus glyceraldehyde-3-phosphate dehydrogenase (GAPDH). *, ** and ^#^ indicate *p* < 0.05, *p* < 0.01, and *p* < 0.001 vs. DOX treated cells. –: 0.2 μM DOX-untreated, +: 0.2 μM DOX-treated (**C**) Cells were attached for 24 h and then co-treatment with 0.2 μM DOX and various concentration of ATG (0–200 μM) for 72 h in DMEM medium supplemented with 2% FBS. The proteins in whole cell lysates were separated by 8% or 15% sodium dodecyl sulfate (SDS)-polyacrylamide gel electrophoresis (PAGE) and transferred to polyvinylidenefluoride (PVDF) membranes. Bands were densitometrically analyzed using Scion Image and band densities were normalized versus β-actin. ^$^ and ^$$^ indicate *p* < 0.01 and *p* < 0.005 vs. non-treated control. *, ** and *** indicate *p* < 0.05, *p* < 0.01 and *p* < 0.005 vs. DOX treated cells. (B, C) All experiments were conducted independently three times and results are presented as means ± SDs.

**Figure 3 ijms-21-02997-f003:**
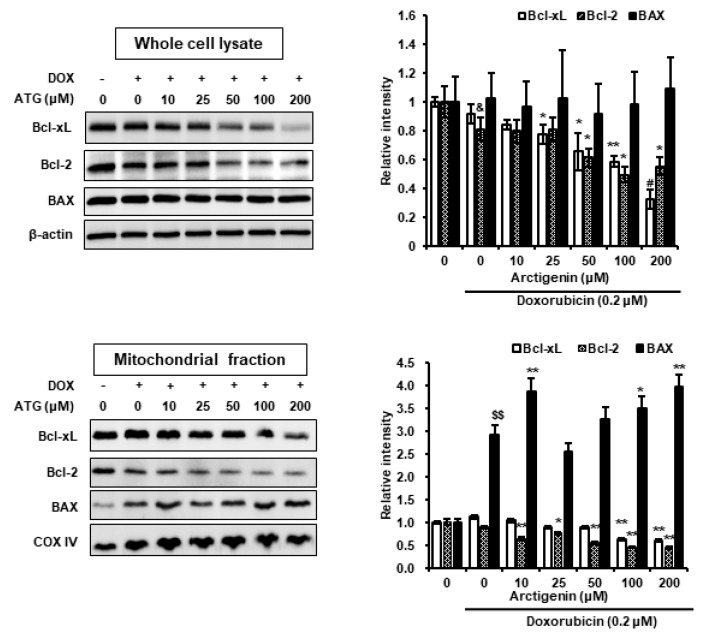
Effects of ATG/DOX co-treatment on mitochondrial Bcl-xL, B-cell lymphoma 2 (Bcl-2), and BAX levels in MDA-MB-231 cells. Cells were attached for 24 h and then co-treated with 0.2 μM DOX and various concentrations of ATG (0–200 μM) for 72 h in DMEM supplemented with 2% FBS. ATG/DOX co-treatment reduced Bcl-xL and Bcl-2 levels in whole cell lysate and mitochondria. In contrast, mitochondrial BAX was increased by ATG/DOX co-treatment. The proteins in whole cell lysates and mitochondrial fractions were separated by 15% SDS-PAGE and transferred to PVDF membranes. β-Actin and mitochondrial cytochrome c oxidase subunit IV (COX IV) were used as internal controls in whole cell lysates and mitochondrial fractions, respectively. All bands were densitometrically analyzed using Scion Image and band densities in samples were normalized versus β-actin and COX IV, respectively. ^&^ and ^$$^ indicate *p* < 0.05 and *p* < 0.005 vs. non-treated controls. *, ** and ^#^ indicate *p* < 0.05, *p* < 0.01, and *p* < 0.001 vs. DOX treated cells. Experiments were conducted independently three times and results are presented as means ± SDs. –: 0.2 μM DOX-untreated, +: 0.2 μM DOX-treated.

**Figure 4 ijms-21-02997-f004:**
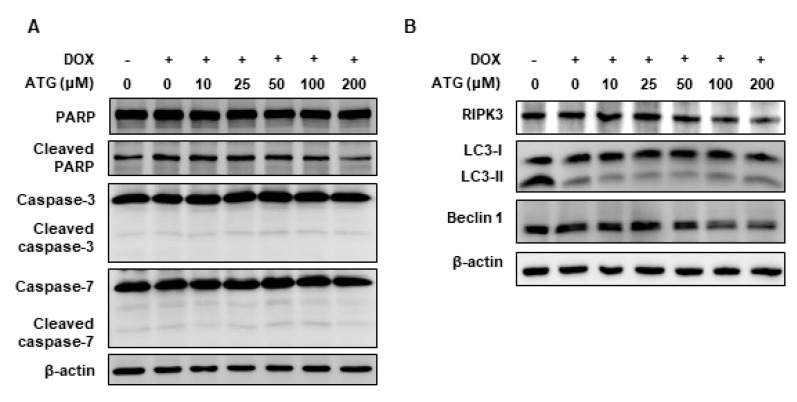
Effects of ATG/DOX co-treatment on the activations of caspase-3 and -7 and on the expressions of necroptosis and autophagy factors in MDA-MB-231 cells. (**A**,**B**) Cells were attached for 24 h and then co-treated with 0.2 μM DOX and various concentration of ATG (0–200 μM) for 72 h in DMEM medium supplemented with 2% FBS. –: 0.2 μM DOX-untreated, +: 0.2 μM DOX-treated (**A**) ATG/DOX co-treatment did not affect the cleavages of poly (ADP-ribose) polymerase (PARP), caspase-3, or caspase-7. Proteins in whole cell lysates were separated by 10% or 15% SDS-PAGE and transferred to PVDF membranes. All experiments were conducted independently three times. (**B**) ATG/DOX co-treatment did not affect the expression of receptor interacting serine/threonine kinase 3 (RIPK3), LC3-II or Beclin. Proteins in whole cell lysates were separated by 8% or 15% SDS-PAGE and transferred to PVDF membranes. All experiments were conducted independently three times.

**Figure 5 ijms-21-02997-f005:**
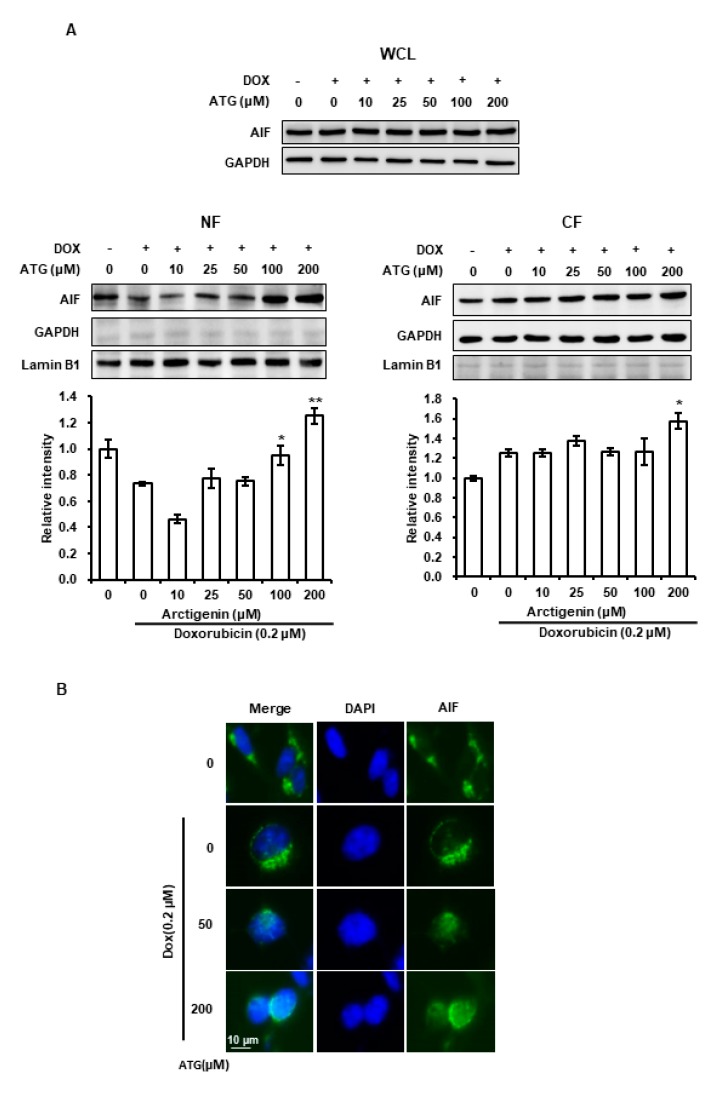
Effects of ATG/DOX co-treatment on the nuclear translocation of apoptosis inducing factor (AIF) in MDA-MB-231 cells. (A,B) Cells were attached for 24 h and then co-treated with 0.2 μM DOX and various concentration of ATG (0–200 μM) for 72 h in DMEM medium supplemented with 2% FBS. (**A**) Nuclear AIF levels were increased by ATG/DOX co-treatment. Proteins in whole cell lysates and in nuclear and cytosolic factions were separated by 10% or 15% SDS-PAGE and transferred to PVDF membranes. Nuclear and cytosolic fraction bands were densitometrically analyzed using Scion Image software and band densities were normalized versus Lamin B1 and GAPDH, respectively. *p* < 0.05 and *p* < 0.005 vs. non-treated controls. * and ** indicate *p* < 0.05 and *p* < 0.01 vs. DOX-only. All experiments were independently conducted three times, and results are presented as means ± SDs. −: 0.2 μM DOX-untreated, +: 0.2 μM DOX-treated (**B**) After serial fixation with ice-cold methanol and acetone, we immunostained for cellular AIF and nuclei were stained with 4’,6-diamidino-2-phenylindole (DPAI) supplemented antifade mounting solution. Cells were visualized under a fluorescence microscope. ATG/DOX co-treatment was found to promote the nuclear translocation of AIF.

**Figure 6 ijms-21-02997-f006:**
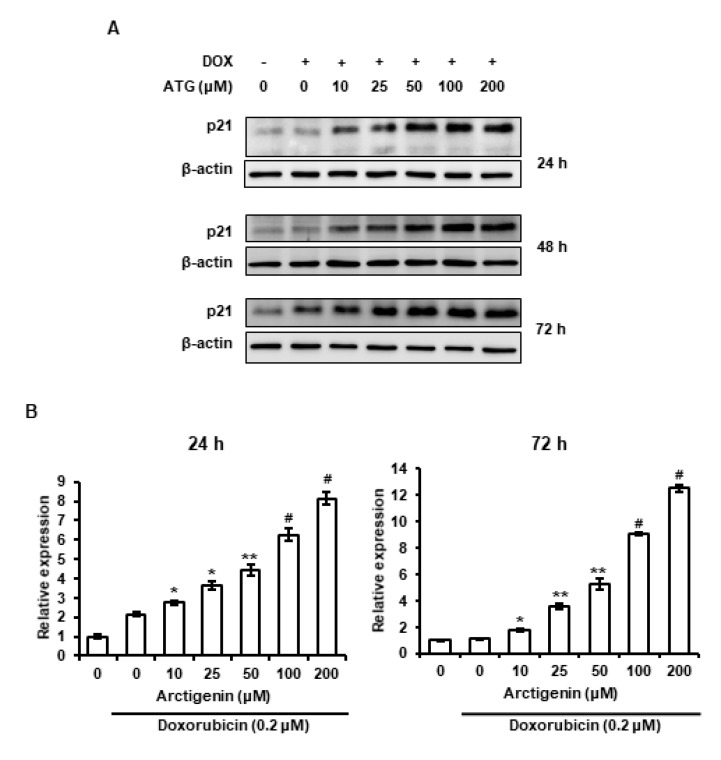
Effects of ATG/DOX co-treatment on p21 expression in MDA-MB-231 cells. (**A**) Cells were attached for 24 h and then co-treated with 0.2 μM DOX and various concentration of ATG (0–200 μM) for 24, 48, or 72 h in DMEM medium supplemented with 2% FBS. The expression of p21 protein was increased by ATG/DOX co-treatment. Proteins in whole cell lysates were separated by 15% SDS-PAGE and transferred to PVDF membranes. –: 0.2 μM DOX-untreated, +: 0.2 μM DOX-treated (**B**) Cells were attached for 24 h and then co-treated with 0.2 μM DOX and various concentration of ATG (0–200 μM) for 24 and 72 h in DMEM supplemented with 2% FBS, respectively. p21 mRNA levels were enhanced by ATG/DOX co-treatment. Relative p21 mRNA expressions were determined in triplicate and normalized versus GAPDH. *, **, ^#^ and ^##^ indicate *p* < 0.05, *p* < 0.01, *p* < 0.001, and *p* < 0.0001 vs. DOX treated cells.

**Figure 7 ijms-21-02997-f007:**
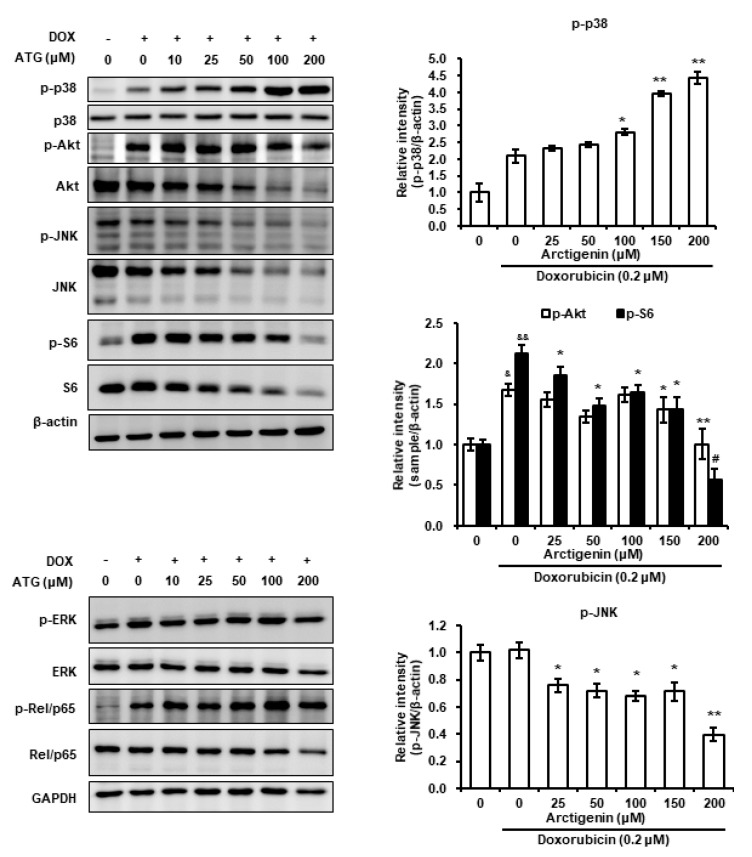
Effects of ATG/DOX co-treatment on the phosphorylations of Akt, mitogen-activated protein kinases (MAPKs), S6 and NF-κB in MDA-MB-231 cells. Cells were attached for 24 h and then co-treated with 0.2 μM DOX and various concentration of ATG (0–200 μM) for 72 h in DMEM medium supplemented with 2% FBS. Phosphorylated p38 levels induced by DOX were concentration-dependently increased by ATG. In contrast, DOX-induced phosphorylation of S6 was prevented by ATG. ATG/DOX co-treatment reduced the phosphorylations of Akt and JNK. Proteins in whole cell lysates, nuclear fractions, and cytosolic factions were separated by 8% or 10% SDS-PAGE and transferred to PVDF membranes. Bands in nuclear and cytosolic fractions were densitometrically analyzed using Scion Image software and band densities were normalized versus GAPDH. ^&^ and ^&&^ indicate *p* < 0.05 and *p* < 0.01 vs. non-treated controls. *, ** and ^#^ indicate *p* < 0.05, *p* < 0.01, and *p* < 0.001 vs. DOX treated cells. All experiments were conducted independently three times, and the results are presented as means ± SDs. –: 0.2 μM DOX-untreated, +: 0.2 μM DOX-treated

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
