# Peer review of "Arctigenin Enhances the Cytotoxic Effect of Doxorubicin in MDA-MB-231 Breast Cancer Cells"

_ijms, 2020, doi:10.3390/ijms21082997_

Round 1
Reviewer 1 Report
The authors Lee et all have described in the following research article the role of Arctigenin in the enhancement of cytotoxic effects in Breast cancer cells. There are a few minor issue that the authors need to address.
- Is this only seen the MB-231 cells or do you see the following effect of the drug in other breast cancer cell lines or cell lines bearing the same mutational landscape of the Mb-231 cells. It would be good to mention on the presence or lack of Arctigenin on other breast cancer cell lines or any other cancer per say where Dox is a primary mode of treatment.
- The standard of care of Breast cancer with Dox is cyclophosphamide or 5-FU. Is there data to support the synergistic effect in combination with these drugs if not why please describe as to why you would think DOx+Arctigenin would be considered as a better therapy.
The authors have otherwise done a considerable amount of work that is appreciated. I would recommend the authors to address the above minor issues deeming the article fit for publication.
Author Response
Thank you for your sharpen indications and suggestions.
Reviewer #1
The authors Lee et all have described in the following research article the role of Arctigenin in the enhancement of cytotoxic effects in Breast cancer cells. There are a few minor issue that the authors need to address.
1. Is this only seen the MB-231 cells or do you see the following effect of the drug in other breast cancer cell lines or cell lines bearing the same mutational landscape of the Mb-231 cells. It would be good to mention on the presence or lack of Arctigenin on other breast cancer cell lines or any other cancer per say where Dox is a primary mode of treatment.
: Thanks for your comment. We investigated the effect in just only MDA-MB-231 breast cancer cells. No papers showed the effects of combinational treatment with DOX and arctigenin in breast cancer cells and other cancer cells. However, anti-cancer activity of arctigenin in MDA-MB-231 and MDA-MB-468 triple-negative breast cancer cells was reported and Wang et al. revealed that cisplatin sensitivity was enhanced by arctigenin in non-small cell lung carcinoma H460 cells. Therefore, we thought arctigenin should increase doxorubicin sensitivity. We mentioned the reports in Discussion section.
2. The standard of care of Breast cancer with Dox is cyclophosphamide or 5-FU. Is there data to support the synergistic effect in combination with these drugs if not why, please describe as to why you would think DOx+Arctigenin would be considered as a better therapy.
: Thanks for your comment. Combined treatment of 5-FU, Doxorubicin and cyclophosphamide (FAC) is a useful strategy to treat breast cancer. However, FAC has many side effects including infection, vomiting, diarrhea and heart damages. In several investigations showed that no significant side effects of arctgenin was oberseved less than 6 mg/kg in vivo (Front. Pharmacol., 16 October 2019 | https://doi.org/10.3389/fphar.2019.01218). Moreover, arctigenin has anti-metastatic potential (Maxwell et al. 2017, International Journal of Oncology). Therefore, we thought combined treatment of Dox and ATG may decrease the side effects and should improve therapeutic efficacy. We mentioned about that in Discussion section.
Reviewer 2 Report
Authors present significant data on how ATG potentiate of the cytotoxic activity of Dox on MDA-MB-231 cells opening doors for a probable combination of ATG/DOX-based chemotherapy with less side effects in cancer therapy. The document is well written. However, few points need to be addressed.
a.
Authors should make effort to discuss limitation to the study and provide perspective to the current work
b.
4.2 Cell culture
Which antimycotic/antibiotic did you use ? Be specific
c.
4.8. Nuclear fractionation
It is not clear to me how you assess the nuclear level of AIF? Please address!
Author Response
Thank you for your sharpen indications and comments. We replied for reviewer’s comments as below:
Authors present significant data on how ATG potentiate of the cytotoxic activity of Dox on MDA-MB-231 cells opening doors for a probable combination of ATG/DOX-based chemotherapy with less side effects in cancer therapy. The document is well written. However, few points need to be addressed.
a.
Authors should make effort to discuss limitation to the study and provide perspective to the current work
: Thanks for your comments. We mentioned the limitation of this investigation and provided perspective to this investigation in Discussion section.
b.
4.2 Cell culture
Which antimycotic/antibiotic did you use? Be specific
: Thanks for your comment. We mentioned components of antimycotic/antibiotic solution in 4.2 Cell culture section.
c.
4.8. Nuclear fractionation
It is not clear to me how you assess the nuclear level of AIF? Please address!
: We assessed the nuclear AIF by Western blotting. To avoid confusion, we mentioned in 4.8 Nuclear fractionation section as “The nuclear fractions were used for assessing the nuclear level of AIF by Western blotting.”